# Probabilistic coherence, logical consistency, and Bayesian learning: Neural language models as epistemic agents

Gregor Betz[1]*, Kyle Richardson[2]

**1** Department of Philosophy, Karlsruhe Institute of Technology, Karlsruhe, Germany, **2** Aristo, Allen Institute for AI, Seattle, WA, United States of America

* gregor.betz@kit.edu

**Data Availability Statement:** The data is now available via wandb under the following links/repos:
https://wandb.ai/doxlm2/doxlm2_model_runs
https://wandb.ai/doxlm2/doxlm2_finetuning
https://wandb.ai/doxlm2/dataset_versions.

## Abstract

It is argued that suitably trained neural language models exhibit key properties of epistemic agency: they hold probabilistically coherent and logically consistent degrees of belief, which they can rationally revise in the face of novel evidence. To this purpose, we conduct computational experiments with RANKERS: T5 models [Raffel et al. 2020] that are pretrained on carefully designed synthetic corpora. Moreover, we introduce a procedure for eliciting a model's degrees of belief, and define numerical metrics that measure the extent to which given degrees of belief violate (probabilistic, logical, and Bayesian) rationality constraints. While pretrained RANKERS are found to suffer from global inconsistency (in agreement with, e.g., [Jang et al. 2021]), we observe that subsequent self-training on auto-generated texts allows RANKERS to gradually obtain a probabilistically coherent belief system that is aligned with logical constraints. In addition, such self-training is found to have a pivotal role in rational evidential learning, too, for it seems to enable RANKERS to propagate a novel evidence item through their belief systems, successively re-adjusting individual degrees of belief. All this, we conclude, confirms the *Rationality Hypothesis*, i.e., the claim that suitable trained NLMs may exhibit advanced rational skills. We suggest that this hypothesis has empirical, yet also normative and conceptual ramifications far beyond the practical linguistic problems NLMs have originally been designed to solve.

## Introduction

Neural language models (NLMs) are powerful natural language processing systems which have sparked a scientific revolution in the field of AI & NLP [1–4] and excel at such diverse tasks as, e.g., machine translation [5], text summarization [6], question answering [7, 8], or natural-language inference [9, 10]. The performance of these systems has exploded with the advent of the so-called Transformer network architecture [11] and has been increasing steadily over the last years (e.g., [12]) through further optimizations of machine learning algorithms and system design, increases in model size, or quantitatively and qualitatively improved training datasets. Technically, and leaving aside all the details, NLMs are essentially **probabilistic**

**Funding:** This work is supported by the Helmholtz Association Initiative and Networking Fund on the HAICORE@KIT partition. The funders had no role in study design, data collection and analysis, decision to publish, or preparation of the manuscript.

**Competing interests:** The authors have declared that no competing interests exist.

**word prediction machines**. They are, first and foremost, trained to fill in missing or next words in a text; and they do predict a word by assigning probabilities to all words available in a given vocabulary.

The strong performance of NLMs in natural language understanding tasks triggers the more fundamental question whether NLMs are rational agents:

(**Rationality Hypothesis**) Suitably designed and trained NLMs may systematically display advanced rational skills.

By discussing the (Rationality Hypothesis), we put the more specific questions addressed in this study (see Q1–Q4 below) in a broader scientific context, sketching their potential relevance for a variety of disciplines and fields.

Rationality is arguably a *contested concept* (like justice). So what exactly does it mean that a NLM possesses advanced rational skills? We take it that such skills would include, more specifically, the abilities to reason correctly (infer, argue, and explain), to produce linguistic output that is sufficiently stable and globally consistent, and to adjust a former output in the light of novel evidence (or, more precisely, a linguistic representation of novel evidence). Moreover, advanced rational behavior of NLMs would allow one to adopt an "intentional stance" [13] towards these systems and to treat them as *doxastic*, if not *epistemic* agents holding beliefs and acquiring knowledge. In this study, we focus on, further specify, and operationalize the aforementioned *epistemic* competences. In doing so, we don't, however, intend to imply that *all* dimensions of rationality can be reduced to such theoretical or epistemic skills.

To say that future NLMs (trained on linguistic data) may exhibit artificial general intelligence (AGI) means to endorse the *Rationality Hypothesis*.

The *Rationality Hypothesis* has ramifications far beyond the practical linguistic problems NLMs have been developed (and are used) to solve. Normatively and conceptually, its investigation may shed new light on the notion of rationality itself (see [14]), helping us to see whether reason is an *emergent property* [15]: Is reliable rational behavior a cognitive macro pattern that *emerges* when agents exercise basic linguistic skill (predicting missing words)? Or, to give this a normative twist: The *Rationality Hypothesis* asks which, if any, rational practices are grounded in elementary language norms. Empirically, an investigation of the *Rationality Hypothesis* will potentially alter our scientific understanding of human cognition (see also [16, 17], especially so as NLMs are found to accurately predict humans' behavioral and neural responses to linguistic stimuli [18–20]. Accordingly, the *Rationality Hypothesis* evokes the possibility that humans may exhibit some kinds of advanced rational competences simply because, and to the extent that they master a suitable language (in the sense of being able to generate sensible texts, basically by next-word prediction). If next-word prediction were all you need for rationality, then language acquisition might be explained without postulating Chomskyan innate linguistic knowledge [21]; then neuro-cognitive theories of the mind could possibly dispense with proper 'rationality modules' [22]; then so-called systems 1 and 2 in dual process theories [23, 24] might simply correspond to different 'modulations' of (i.e., ways of exercising) an essentially linguistic cognitive capacity; and then the evolution of reason (as, for instance, described by [25]) might proceed entirely through an evolution of language usage, and without changes in the underlying neuro-cognitive mechanisms and their genetic basis. All in all, these considerations stress the significance and fruitfulness of the *Rationality Hypothesis*.

The breathtaking performance of NLMs, pro-active science communication (e.g., surrounding GPT-2), and the increasing use of powerful state-of-the-art NLMs in production environments have raised public attention and provoked a lively debate about the nature of these systems. The popular and scientific hopes associated with NLMs have been denounced as hype [26], as yet another wave of overly-optimistic AI dreams [27]. NLMs' strong performance, it

has been argued, might be due to memorization effects (quantified more thoroughly by [28, 29]), or hidden statistical cues in the datasets [30, 31]. And NLMs have been exposed by making them produce nonsensical output [26, 32]. So, doesn't this falsify the *Rationality Hypothesis*, and thus settle the question whether NLMs are rational agents? Not at all. Note, first of all, that the *Rationality Hypothesis* is an existential statement. Hence, on purely logical grounds, it cannot be falsified by testing a specific NLM. And even if all current NLMs were shown to fail systematically, this would at best represent inductive (and hence defeasible) evidence against the hypothesis. Secondly, while the critical debate has pointed out methodological pitfalls to avoid and important shortcomings of current NLMs (and we will turn to these below), the specific assessments of NLMs sometimes lack systematic rigour (as, e.g., in [32]), and are therefore inconclusive.

So, what exactly is the evidence for or against the *Rationality Hypothesis*? Systematic evaluations reveal that some NLMs display significant zero-shot performance [4] and transfer learning ability [33, 34], all of which speaks in favor of the *Rationality Hypothesis*. Likewise, NLMs can be specifically trained (i.e., fine-tuned) to master diverse inferential and explanatory reasoning tasks, such as natural language inference [10], deductive argumentation [35, 36], enthymematic and abductive inference [37], defeasible reasoning [38], rule-based planning [39], explaining answers in QA tasks [40], proof generation in natural language [41, 42] and mathematics [43–45], or argument analysis [46]. However, despite the astonishing and wide-ranging successes of NLMs, studies that have carefully probed current systems equally point out major limitations that are pertinent to the *Rationality Hypothesis*, namely: NLMs' sheer **inability to produce coherent and consistent output** [47, 48], along with a failure to properly handle negations [49–51]. From this evidence, we must conclude that *current* pre-trained NLMs are *not* doxastic, let alone epistemic agents.

In response to NLMs' apparent lack of coherence and consistency, projects that seek to build artificial rational agents have, recently, started to embed NLMs as sub-parts in modular cognitive systems which contain further components, e.g., for monitoring the NLM and enforcing output consistency relative to domain-specific constraints. Accordingly, [52] outline, in general terms, an architecture with persistent memory and language models, orchestrated by a central controller. Following this framework, NLMs have been paired with constraint-enforcing modules so as to build systems for causal inference [53], for knowledge representation [54, 55], and for robust story generation [56]. Such an approach parallels dual-process theories [23], equating NLMs with fast yet error-prone system 1, which is complemented by a rule-based system 2. While being promising steps towards AGI, and representing significant engineering projects in their own, these studies are only of limited relevance for probing the *Rationality Hypothesis*: rationality is *explicitly built* into these systems via apriori constraints. Therefore, the performance of such modular systems is neither evidence for nor against the hypothesis that rational reasoning abilities may emerge from language skills alone.

Multi-modal coupling, where linguistic data is processed alongside sensorical data, represents an alternative way for embedding a NLM in a larger system. Here, the NLM is, e.g., coupled with an image processing net—rather than with a constraint-enforcing module, as in dual-process approaches. In a programmatic article, [57] argue that AI systems will only master the full spectrum of human language usage if AI language learning is "grounded" in senso-experiential, interactive, ultimately social environments. This represents an intriguing research programme, and we may note: If meaning consists in proper language use [58], and if, for example, it belongs to the function of an evidential statement that it may register a sensory experience of an agent, or guide an agent's course of action (intervention) which allows her to probe the very statement, then, clearly, word prediction machines trained on linguistic data only cannot fully "grasp" the meaning of an evidential statement. But to say so does not refute

the *Rationality Hypothesis* (and [57] never claimed so). The interrelation between extra-linguistic grounding and an AI system's ability to exercise *rational* linguistic faculties is not settled by the philosophical arguments [57] set forth, and it represents a pertinent research topic in its own. In this regard, [59] seem to provide evidence that visual grounding increases the global logical consistency of an AI system's linguistic output: Accordingly, grounding may very well be a suitable means for improving rational reasoning in NLMs. However, is grounding also necessary for acquiring the competencies of sound reasoning, consistent belief formation, or rational belief revision? (There are philosophical reasons to doubt this: If perceptual grounding were necessary for rational reasoning skill, then the whole notion of apriori rationality, i.e., the idea that at least some normative principles of correct reasoning, such as deductive inference, hold independently of any empirical facts and knowledge thereof, risks to be in limbo. Because it would be puzzling if some apriori truth can *only* become to be known by means of empirical learning—or wouldn't it? To say that empirical experience is necessarily required for mastering a certain rational faculty implies that norms which govern this faculty are not analytic truths, but rather represent—to use a venerable terminology—synthetic apriori statements.) This study, for sure, suggests otherwise: We present simple and pure NLMs which, while being trained on synthetic text corpora alone, show clear signs of advanced rational behavior.

This paper diagnoses and studies remedies for the poor consistency of current pre-trained NLMs without suggesting to embed NLMs in a broader cognitive architecture, as in grounding or dual-process approaches. It rests on three basic working assumptions: First, given that NLMs are *probabilistic* word prediction machines, Bayesian epistemology [60]—broadly construed—seems to be a suitable normative framework to conceptualize and probe NLMs' rationality. Second, a NLM may form a stable, sufficiently consistent belief system through "self-training," i.e., training on coherent texts which the NLM has generated itself and which express its provisional beliefs. Third, the laws and mechanisms of belief formation in NLMs can (so far) best be studied in a fully-controlled experimental set-up, where NLMs are trained from scratch on synthetic corpora in a simple, artificial language (we will resort to a language for expressing linear orders). In these two latter regards, the paper follows [61].

More precisely, we experimentally test the *Rationality Hypothesis* by means of specific NLMs, which we shall call RANKERS. RANKERS instantiate a T5 base configuration [1] and are trained on an artificial language for expressing strict orders. We create an entire ensemble of RANKERS by training on various, carefully created synthetic corpora. This allows us to fully control the pre-training data and to define a reliable procedure for *eliciting the degrees of belief* (credences) of a RANKER. Thus equipped, we address four main questions:

**Q1.** Are the degrees of belief elicited from a pre-trained RANKER both probabilistically coherent and logically consistent (i.e., aligned with the language's logico-semantic constraints), provided that the RANKER itself has been trained on consistent texts?

**Q2.** Does self-training improve probabilistic coherence and logical consistency of degrees of belief?

**Q3.** Can RANKERS successfully integrate novel evidence into their belief system without loosing probabilistic coherence and logical consistency?

**Q4.** Do RANKERS adjust their belief state given novel evidence in accordance with Bayesian learning (conditionalization)?

The results of our computational experiments appear to refute Q1, but represent evidence for affirming Q2, Q3, and Q4. RANKERS, that is, can be considered as epistemic agents

that hold a probabilistic belief system and rationally adjust their credences in light of novel evidence. While the *Rationality Hypothesis*, for being an existential statement, cannot be *falsified* by a single example, as noted before, it may very well be *verified* by a suitable instance. That's how we conceive—generally speaking—of this study: It develops and assesses NLMs which, while being purely trained as word prediction machines, reliably display, to a certain extent, advanced rational behavior, and hence confirm the *Rationality Hypothesis*.

The paper proceeds as follows. Section Formal approach develops the formal frameworks used in the computational experiments, in particular: the simple artificial language RANKERS will be trained on (Artificial language); the procedure for eliciting conditional degrees of belief from a RANKER (Conditional elicitation); a global measure of a belief system's informational content (Doxastic entropy); as well as the rationality constraints on degrees of belief—and corresponding numerical metrics—pertaining to probabilistic coherence (Popper metrics), logical consistency (Consistency metrics), and evidential learning (Bayesian metrics). Section Design of Computational Experiments describes the design of our experiments, which are composed of three consecutive phases: In PHASE 1, RANKERS are pre-trained on carefully designed synthetic corpora (Corpus construction and Pre-training); in PHASE 2, the pre-trained RANKERS further train on auto-generated texts (Self-training); finally, in PHASE 3, the self-trained RANKERS are exposed to novel evidence according to different so-called evidence introduction regimes (Evidential learning). Section Results presents our major empirical findings: Self-training improves the probabilistic coherence and logical consistency of language models as measured by the corresponding metrics (Pre evidence introduction (PHASE 2). In addition, continuously self-training RANKERS are able to propagate novel evidence through their belief system in rough agreement with rational constraints on evidential learning (Post evidence introduction (PHASE 3). Ablation studies, which turn off self-training, demonstrate its pivotal role during evidential learning for maintaining a globally coherent belief system (Ablation studies). Section Discussion concludes with a brief summary, a discussion of limitations, and directions of future research.

## Formal approach

### Artificial language

The artificial language described in the following, which has been used in computational experiments with NLMs before [61], unites efficient semantics with a sufficiently rich inferential structure. In particular, it has all the logical properties required to give rise to voting paradoxes in judgment aggregation [62], which, it has been argued [61], is a potential reason for why pretrained NLMs are prone to suffer from inconsistencies.

Let us first describe the syntax and then the simple semantics of the artificial language $L$. $L$'s alphabet consists of $k$ constants $a_1 \ldots a_k$ (we set $k = 200$ in our experiments), and two binary predicates $R$, $S$. A $L$-sentence has the form $xXy$ (with $x$, $y$ and $X$ being terms in the meta-language referring to constants, respectively predicates in $L$). $L$ just contains such "atomic" sentences. The following rules define the logic of $L$ (clearly, these rules are themselves not expressible in $L$):

**Irreflexivity.** For any constant $a_i$: $a_iRa_i \vdash \perp$.

**Asymmetry.** For any constants $a_i$, $a_j$ ($i \neq j$): $a_iRa_j, a_jRa_i \vdash \perp$.

**Duality.** For any constants $a_i$, $a_j$ ($i \neq j$): $a_iRa_j \dashv\vdash a_jSa_i$.

**Transitivity.** For any (pairwise non-identical) constants $a_i$, $a_j$, $a_l$: $a_iRa_j, a_jRa_l \vdash a_iRa_l$.

A set of $L$-sentences is consistent iff one cannot deduce $\bot$ by iteratively applying these rules.

The semantics of this language are straightforward. Consider a domain with $k$ objects, mapped bijectively to $L$'s constants by interpretation $I$, then every total strict order of these objects represents a possible world $w$, with $R$ ($S$) corresponding to the order relation "$<$" ("$>$"). A $L$-sentence $\varphi$ (of form $a_iRa_j$, or $a_iSa_j$) is true in the possible world $w$ with order $<_w$, in short $\vDash_w \varphi$, iff $I(a_i)<_w I(a_j)$, resp. $I(a_i)>_w I(a_j)$. We write $\vDash_w A$ with $A \subset L$ iff $\vDash_w \varphi$ for all $\varphi \in A$. A set of $L$-sentences $A$ is consistent iff there exists a possible world $w$ such that all sentences in $A$ are true in $w$, i.e. $\vDash_w A$. The proposition expressed by a set of $L$-sentences $A$ is the set of all possible worlds that model $A$, which can in turn be identified with a (typically non-total) strict order on the domain. $L$-sentences $A$ entail $L$-sentence $\varphi$, $A \vDash \varphi$, iff $\varphi$ is true in every possible world $w$ with $\vDash_w A$.

Let us establish, for convenience, the following conventions:

We write $L$ for the set of all $L$-sentences.

For some $L$-sentence $\varphi$ (of form $a_iRa_j$, or $a_iSa_j$), we obtain $\bar{\varphi}$ by exchanging its predicate (replacing $R$ with $S$, or vice versa); given $L$'s semantics, $\bar{\varphi}$ is the **negation** of $\varphi$. Similarly, we obtain $\tilde{\varphi}$, the **equivalent** of $\varphi$, by exchanging its predicate and swapping the order of its two constants (e.g., $a_iRa_j$ is equivalent to $a_jSa_i$).

We write $S_{all}$ for the set of all **finite sequences** of $L$-sentences, including the empty sequence. $S_{all}$ is closed with respect to concatenation, i.e., if $x \in S_{all}$ and $y \in S_{all}$, then $xy \in S_{all}$. The set of all $L$-sentences in some sequence $x \in S_{all}$ shall be referred to as $\|x\|$. Specific subsets of $S_{all}$ are:

- $S \subset S_{all}$: the set of all **consistent** finite sequences of $L$-sentences;

- $S_\varphi \subset S$: the set of all consistent finite sequences of $L$-sentences that contain some sentence $\varphi \in L$;

- $S_A \subset S$: the set of all consistent finite sequences of $L$-sentences that contain every sentence $\varphi \in A$ for some $A \subseteq L$.

Finally, let $x|_{\varphi, \psi} \in S$ be the sequence that is obtained from sequence $x \in S$ by replacing every occurrence of $\varphi$ in $x$ with $\psi$.

## Conditional elicitation

Informally speaking, we equate a RANKER's degree of belief in some statement $a_iRa_j$ with its propensity to predict that the missing token in $a_i$ [mask] $a_j$ is $R$.

Let $M$ be a neural language model capable of masked token prediction (i.e., probabilistic missing-word prediction) in our language $L$. To elicit the model's conditional credence in a $L$-sentence $a_iRa_j$ (likewise $a_iSa_j$) given a sequence of $L$-sentences $x$, we concatenate $x$ and $a_iRa_j$, mask the predicate letter, which yields $x\, a_i$ [mask] $a_j$, query the model, and interpret the model's probability prediction for [mask] $= R$, the so-called confidence, as its conditional degree of belief in $a_iRa_j$ given $x$, in short:

$$Bel_M(a_iRa_j|x) = \text{Prob}_M([\text{mask}] = R|x\, a_i\, [\text{mask}]\, a_j).$$

## Doxastic entropy

Assuming that the credences of model $M$, $Bel_M(\cdot)$, represent a joint probability distribution on all $L$-sentences, we may define $M$'s **doxastic entropy** as the credences' Shannon entropy, $H(Bel_M)$.

As we will see later on, credences are not perfect probability functions. However, we may measure the doxastic entropy of a model's belief system nonetheless by eliciting absolute credences for a random sample $X \subset L$ of $L$-sentences ($N = 1024$) and calculating the mean *joint* entropy of these $N$ different probability functions (which we treat as independent), i.e.

$$H_M = \frac{1}{|X|} \sum_{\varphi \in X} H(Bel_M(\varphi))$$

with

$$H(Bel_M(\varphi)) = -Bel_M(\varphi) \cdot \log_2(Bel_M(\varphi)) - (1 - Bel_M(\varphi)) \cdot \log_2(1 - Bel_M(\varphi)).$$

Doxastic entropy measures, simply put, the global informativeness of a belief system; it is minimal iff all credences are either 1 or 0, and maximal iff all credences equal .5 (zero information content).

In Section Results, when analyzing the experimental results, we will distinguish and group RANKER models according to their mean doxastic entropy during self-training (average PHASE 2-entropy).

## Popper metrics

Probabilities are typically defined and studied in a formal *semantic* framework, i.e. with reference to a given *event* space, or an algebra of *propositions*. Language models are, however, basically syntactic systems, processing and manipulating token sequences. In this paper's context, in particular, degrees of belief are elicited for $L$-sentences (see Conditional elicitation), rather than propositions. To assess the probabilistic coherence of a NLM's degrees of belief, we therefore resort to a sentential framework of probability, proposed by Popper [63]. In this framework, conditional probabilities are introduced as a real-valued function from pairs of sequences ($\mathcal{S} \times \mathcal{S} \to \mathbb{R}$). Popper assumes that two operations—complement and conjunction—are defined on the underlying set of sequences $\mathcal{S}$, and that $\mathcal{S}$ is closed with respect to both, i.e., if $\varphi \in \mathcal{S}$, then $\bar{\varphi} \in \mathcal{S}$, and if, in addition, $\psi \in \mathcal{S}$, then $\varphi \wedge \psi \in \mathcal{S}$.

Now, the straightforward application of Popper's framework to our case is to interpret $\mathcal{S}$ as the set of all (finite) sequences of $L$-sentences ($\mathcal{S} = S_{all}$), to define the operation of conjunction as concatenation, and to equate complement with syntactic negation. Yet, with this natural interpretation, the operation of complement on $\mathcal{S}$ is not closed: Our artificial language only allows for expressing the negation of individual sentences, and doesn't provide resources to directly express the negation of a sequence of $L$-sentences. Negation is restricted to atomic sequences in $S_{all}$. This necessitates a first deviation from Popper's framework: Probabilistic constraints that refer to negation will have to be restricted to individual sentences, i.e. *atomic* elements of $S_{all}$.

[63] discusses different versions of his system of probability. Our starting point is the following set of axioms (adapted from [64]):

**Definition relative probability.** A real-valued function $Pr : \mathcal{S} \times \mathcal{S} \to [0; 1]$ is called a relative probability function (or, Popper measure), iff the following constraints hold for arbitrary elements $\varphi, \psi, \chi \in \mathcal{S}$:

$$Pr(\varphi|\varphi) = 1 \tag{Reflexivity}$$

$$Pr(\varphi\psi|\chi) = Pr(\psi\varphi|\chi) \tag{Commutation}$$

$$Pr(\varphi|\psi\chi) = Pr(\varphi|\chi\psi) \tag{Commutation}$$

$$Pr(\varphi|\psi) + Pr(\bar{\varphi}|\psi) = 1 \qquad \text{(Complement)}$$

$$Pr(\varphi\psi|\chi) = Pr(\varphi|\psi\chi) \times Pr(\psi|\chi) \qquad \text{(Multiplication)}$$

We can now see a second problem with applying Popper's framework to our case: Our elicitation procedure is only defined for individual $L$-sentences, and not for sequences thereof. The expression $Bel(\varphi\psi|\chi)$ (with $\varphi, \psi, \chi$ being $L$-sentences) has simply no meaning. Rather than testing for (Multiplication*) and (Commutation*), we therefore check whether degrees of belief satisfy a weaker condition, namely the following *implication* of those two constraints (see also S1 Appendix):

$$Pr(\varphi|\psi\chi) \times Pr(\psi|\chi) = Pr(\psi|\varphi\chi) \times Pr(\varphi|\chi) \qquad \text{(Multiplication)}$$

For each of the resulting four probabilistic constraints—(Reflexivity), (Commutation), (Complement), (Multiplication)—, the corresponding **Popper metric** measures the extent to which given degrees of belief violate the constraint. More specifically, given credence function $Bel(\cdot|\cdot)$, the Popper metric for a *constraint* equals the mean square difference (MSD) between the left-hand side and the ride-hand side of the equation that expresses the constraint. Formally, for some $X \subseteq L \times L \times S$,

$$\textsc{PopperReflx}(X) = \frac{1}{|X|} \sum_{\langle\varphi,\psi,\chi\rangle \in X} \left(1 - Bel(\varphi|\varphi)\right)^2$$

$$\textsc{PopperCommu}(X) = \frac{1}{|X|} \sum_{\langle\varphi,\psi,\chi\rangle \in X} \left(Bel(\varphi|\psi\chi) - Bel(\varphi|\chi\psi)\right)^2$$

$$\textsc{PopperCompl}(X) = \frac{1}{|X|} \sum_{\langle\varphi,\psi,\chi\rangle \in X} \left(1 - Bel(\varphi|\chi) - Bel(\bar{\varphi}|\chi)\right)^2$$

$$\textsc{PopperMltpl}(X) = \frac{1}{|X|} \sum_{\langle\varphi,\psi,\chi\rangle \in X} \left(Bel(\varphi|\psi\chi) \times Bel(\psi|\chi) - Bel(\psi|\varphi\chi) \times Bel(\varphi|\chi)\right)^2$$

In order to assess the probabilistic coherence of a RANKER model at a given step t, we randomly sample 1024 triples $\langle\varphi, \psi, \chi\rangle \in L \times L \times S$, containing two atomic sequences $\varphi, \psi$ and a consistent sequence of $L$-sentences $\chi$ with maximum length 3. We elicit, for each triple in $X$, all conditional and absolute degrees of belief referenced in the equations above, and finally calculate the four Popper metrics.

The Popper metrics can be considered as computationally handable proxies for a more systematic and conceptually compelling measure of global incoherence, namely the distance between actual credences $Bel(\cdot|\cdot)$ and the closest, probabilistically coherent credence function $Pr_{Bel}(\cdot|\cdot)$ (see also [65, 66] and S2 Appendix).

## Consistency metrics

Consistency metrics are supposed to measure the extent to which degrees of belief respect the objective inferential relations which hold between the sentences in a domain. The artificial language $L$ contains two basic kinds of inferential relations: equivalence and transitivity. This translates into the following logical constraints for degrees of belief on $L$. If sentence $\varphi$ is

entailed by sentences $P$, then the conditional degree of belief in $\varphi$ given $P$ is 1. Any two equivalent sentences $\varphi, \tilde{\varphi}$ are believed to the same degree, and exchanging these sentences in an antecedent condition doesn't alter the corresponding conditional degree of belief, either. Formally:

**Definition logical alignment.**   The degrees of belief $Bel(\cdot|\cdot)$ on $L$ are logically aligned iff the following constraints are satisfied:

$$Bel(\varphi|x_P) = 1 \qquad\qquad\qquad (\text{Entailment})$$

$$Bel(\psi|y) = Bel(\tilde{\psi}|y) \qquad\qquad\qquad (\text{Equivalence})$$

$$Bel(\chi|z_\psi) = Bel(\chi|[z_\psi|_{\psi,\tilde{\psi}}]) \qquad\qquad\qquad (\text{A} - \text{Equivalence})$$

for arbitrary $L$-sentences $\varphi, \psi, \chi \in L$ with $Bel(\varphi) \neq 0$, for any consistent set of $L$-sentences $P \subset L$ such that $\varphi$ follows from $P$ with $L$'s inference rules (possibly requiring multi-hop inference) and corresponding $L$-sequence $x_P \in S$ (i.e., $\|x_P\| = P$), and for any consistent $L$-sequences $y, z_\psi \in S$.

For each logical alignment constraint, the corresponding **consistency metric** measures the extent to which given degrees of belief $Bel(\cdot|\cdot)$ violate the constraint. More specifically, given suitable degrees of belief, the consistency metric for a constraint is the mean square error (MSE) between the left-hand side and the ride-hand side of the equation that expresses the constraint. Formally,

$$\text{LOGALGNENT}(X) = \frac{1}{|X|} \sum_{\langle \varphi, x_P \rangle \in X} \left(1 - Bel(\varphi|x_P)\right)^2$$

for some sample $X \subseteq \{\langle \varphi, x_P \rangle : \varphi \in L \wedge x_P \in S \wedge P \subset L \wedge P \vDash \varphi\}$; and

$$\text{LOGALGNEQV}(X) = \frac{1}{|X|} \sum_{\langle \varphi, x \rangle \in X} \left(Bel(\varphi|x) - Bel(\tilde{\varphi}|x)\right)^2$$

for some sample $X \subseteq \{\langle \varphi, x \rangle : \varphi \in L \wedge x \in S\}$; and

$$\text{LOGALGNEQVA}(X) = \frac{1}{|X|} \sum_{\langle \varphi, x_\psi \rangle \in X} \left(Bel(\varphi|x_\psi) - Bel(\varphi \mid [x_\psi|_{\psi,\tilde{\psi}}])\right)^2$$

for some sample $X \subseteq \{\langle \varphi, x_\psi \rangle : \varphi, \psi \in L \wedge x_\psi \in S\}$.

In applying these consistency metrics to evaluate a RANKER, we restrict the sample ($N = 1024$) to specific sequences; in particular, regarding (Entailment), we only consider sequences $x_P$ with maximum length 5 (2 premises and up to 3 distractors); regarding (Equivalence) and (A-Equivalence), we only consider sequences $x$, resp. $x_\psi$, with maximum length 4.

Moreover, if $L$-sentences $\varphi, \psi$ entail $L$-sentence $\chi$ by transitivity, i.e. $\varphi, \psi \vDash \chi$, we may require that

$$Bel(\chi) \geq Bel(\varphi) \cdot Bel(\psi), \qquad\qquad\qquad (\text{Transitivity})$$

which can be motivated with the probabilistic multiplication rule, and effectively expresses a fuzzy product t-norm (see also S3 Appendix). Conceptually, (Transitivity) is however not a *probabilistic* constraint of credences, but rather represents an additional *objective* constraint on probabilistic degrees of belief.

We define two further consistency metrics that measure, specifically, the extent to which (Transitivity) is violated given some sample $X = \{\langle \varphi, \psi, \chi \rangle \subseteq L \times L \times L : \varphi, \psi \vDash \chi\}$: (i) the relative

frequency of triples in $X$ which violate (Transitivity); (ii) a MSE metric, formally:

$$\text{TransViolRatio}(X) = \frac{|X_\nu|}{|X|}$$

$$\text{TransViolMSE}(X) = \frac{1}{|X|} \sum_{\langle \varphi, \psi, \chi \rangle \in X_\nu} \left(Bel(\chi) - Bel(\varphi) \cdot Bel(\psi)\right)^2$$

with $X_\nu \subseteq \{\langle \varphi, \psi, \chi \rangle \in X : Bel(\chi) < Bel(\varphi) \cdot Bel(\psi)\}$.

We sample 1024 triples of $L$-sentences such that two entail the remaining one in order to assess RANKERS with respect to the two transitivity metrics.

## Bayesian metrics

*Bayesian metrics* are supposed to measure whether an agent adequately adjusts her degrees of belief when being exposed to novel evidence $E$. These metrics assess, in particular, the agent's posterior credences $Pr_{post}$, held after the introduction of novel evidence, in comparison to her credences prior to the exposition to evidence $E$, $Pr_{prior}$. We assume that the novel evidence $E$ is not uncertain, in other words, the model learns, being exposed to the evidence, that $E$ is a fact.

First, a Bayesian agent who learns that $E$ is the case should adjust her posterior beliefs such that the conditional credence in some sentence $\varphi$ given evidence $E$ is equal to the absolute degree of belief in $\varphi$,

$$Pr_{post}(\varphi) = Pr_{post}(\varphi|E) \quad \forall \varphi \in L. \tag{SynchronicLearning}$$

The constraint (Synchronic Learning) is relatively weak, it actually follows from the assumptions that learning evidence $E$ implies $Pr_{post}(E) = 1$ and that absolute credence equals conditional credence relative to a certain event (i.e., an event believed with certainty).

Simple conditionalization amounts to a second, much stronger constraint. Accordingly, agents who learn evidence $E$ are supposed to update their degrees of belief in view of their prior conditional credences relative to the evidence,

$$Pr_{post}(\varphi) = Pr_{prior}(\varphi|E) \quad \forall \varphi \in L. \tag{Conditionalization}$$

Simple conditionalization is a key—arguably: essential—tenet of standard Bayesian epistemology and Bayesian theories of evidential learning. Unlike all previously discussed constraints, Bayesian updating through conditionalization represents—by relating credences held at different points in time—a *diachronic* constraint. Conditionalization regulates the shift from one credence function to another, rather than the internal coherence of a single credence function. The justification of (Conditionalization) as a normative diachronic rationality constraint for Bayesian learning requires further, more demanding arguments than the justification of synchronic probabilistic constraints on credences [67].

(Conditionalization) entails that the conditional credences relative to the evidence $E$ remain the same while updating:

$$Pr_{post}(\varphi|E) = Pr_{prior}(\varphi|E) \quad \forall \varphi \in L. \tag{Likelihoods}$$

(Conditionalization) and (Likelihoods) immediately imply (Synchronic Learning). But (Synchronic Learning) may be satisfied while violating (Likelihoods), for example by keeping prior absolute credences fix and updating conditional credences to prior absolute degrees of belief (setting $Pr_{post}(\varphi|E) = Pr_{prior}(\varphi)$), which may reflect a revision in one's prior degree of

belief in *E*. (To clarify the normative strength of these constraints, we note that the latter amounts to a strategy deemed permissible by non-Bayesian epistemologists in the tradition of, for example, [68–71]; it is, however, ruled out by Bayesians.)

We shall now describe the **Bayesian metrics** we use to track whether a RANKER's degrees of belief conform with the above constraints. We sample, first of all, a fixed set *X* of *L*-sentences (N = 1024). At each self-training step *t*, before and after evidence introduction, we elicit the RANKER model's absolute and conditional degrees of belief, $Bel_t(\varphi)$ and $Bel_t(\varphi|E)$, for all $\varphi$ in *X*. We obtain $Bel_{prior}(\varphi)$ as the average credence during the immediate run-up ($\Delta = 50$) to the evidence introduction at step $t_e$,

$$Bel_{prior}(\cdot|E) = \operatorname*{mean}_{(t_e-\Delta)<t<t_e} (Bel_t(\cdot|E)).$$

Treating all sentences in sample *X* as independent probabilistic variables, we may track the violations of the Bayesian constraints by means of joint relative entropy (Kullback-Leibler divergence).

$$\mathrm{KLsyn}(t) = \frac{1}{|X|} \; KL( \; Bel_t(\cdot) \; || \; Bel_t(\cdot|E) \; )$$

$$\mathrm{KLdia}(t) = \frac{1}{|X|} \; KL\Big( \; Bel_t(\cdot) \; || \; Bel_{prior}(\cdot|E) \; \Big)$$

$$\mathrm{KLcond}(t) = \frac{1}{|X|} \; KL\Big( \; Bel_t(\cdot|E) \; || \; Bel_{prior}(\cdot|E) \; \Big)$$

For an ideal Bayesian learner, all three metrics will become infinitely small after the introduction of novel evidence *E*. Plotting these time-dependent metrics will allow us to obtain detailed insights into how RANKERS adjust their credences, and whether they do so in accordance with Bayesian constraints.

If $Bel_{pre}(\varphi) \approx Bel_{pre}(\varphi|E)$ for some $\varphi \in L$, the novel evidence *E* is of limited significance for some $\varphi$, and Bayesian constraints can trivially be satisfied by not changing the belief in $\varphi$ too much. That's the reason why we focus in the analysis of evidential learning (Section Evidential learning) on statements $\varphi$ in the sample *X* such that $Bel_{pre}(\varphi)$ substantially differs from $Bel_{pre}(\varphi|E)$. More technically, we filter the sample *X* according to the *q*-percentile of $\mathrm{KLsyn}_{prior}$ in sample *X*, with

$$\mathrm{KLsyn}_{prior}(\cdot|E) = \operatorname*{mean}_{(t_e-\Delta)<t<t_e} (\mathrm{KLsyn}_t(\cdot|E))$$

and

$$X' = \{\varphi \in X : \mathrm{KLsyn}_{prior}(\varphi|E) > x\}$$

for some threshold *x* such that $|X'|/|X| = q$.

## Design of computational experiments

### Overview

The overall experimental design is illustrated in Fig 1. We generate 60 synthetic text corpora with the artificial language *L* (Section Artificial language) by *simulating authors* who produce texts which express their consistent belief states (cf. Section Corpus construction). On each corpus, a randomly initialized T5 model [1] is trained on denoising and text-completion tasks

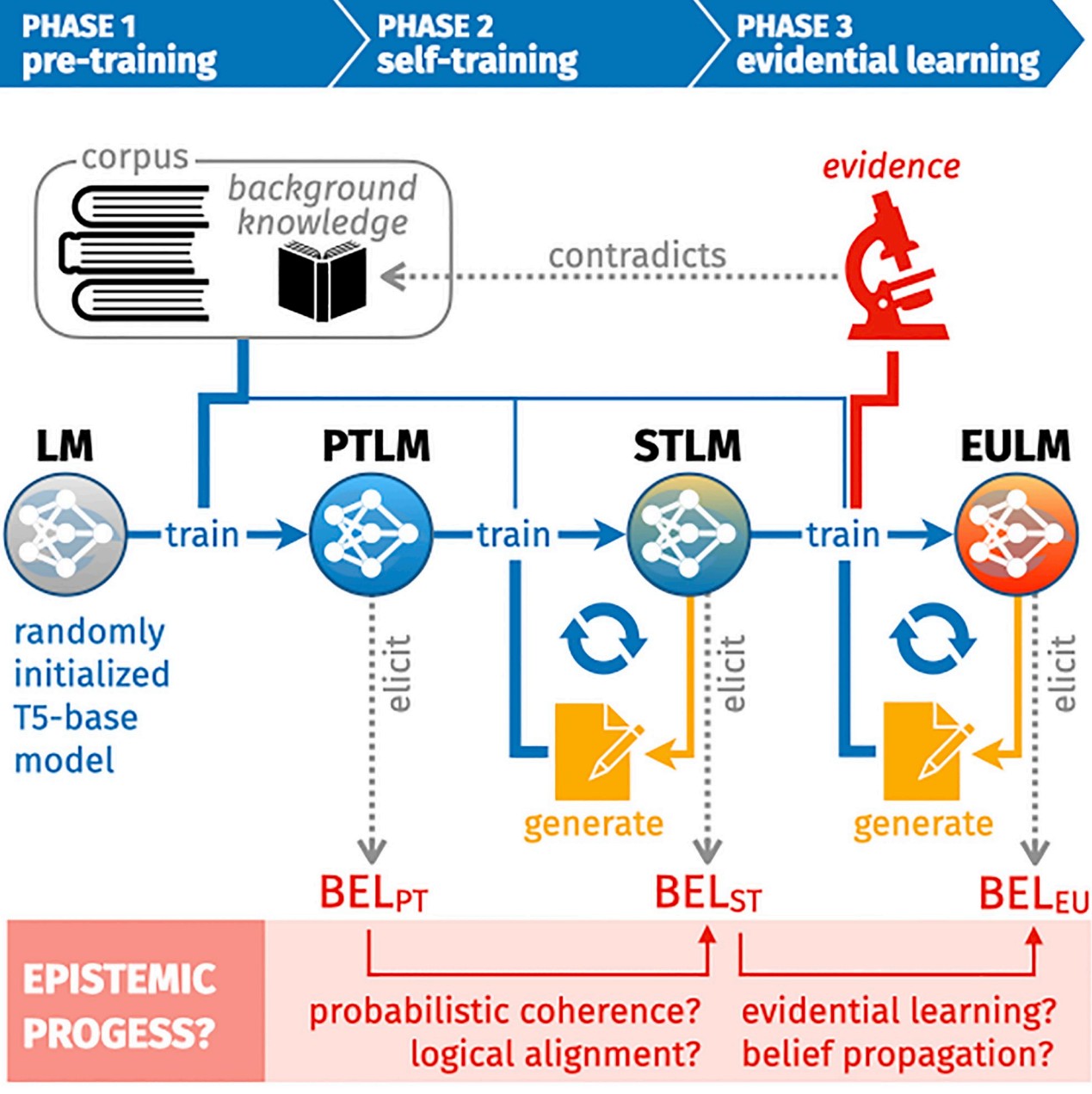

**Fig 1. Overall design.** A randomly initialized RANKER model (LM) is trained on a synthetic text corpus produced by artificial authors. The accordingly pre-trained RANKER model (PTLM) subsequently trains on auto-generated texts, yielding a self-trained RANKER (STLM), which is eventually exposed to novel evidence during training, resulting in an evidentially updated model (EULM).

(cf. Section Pre-training), yielding 60 pre-trained RANKERS at the end of PHASE 1. In PHASE 2, each pre-trained RANKER is submitted to four independent self-training treatments, during which the model continuously generates texts on which it trains instantly (cf. Section Self-training). In PHASE 3, the 240 self-trained RANKER models are exposed to an evidence statement that contradicts a shared assumption of the corresponding, original pre-training corpus. We define two alternative ways for integrating this evidence item into the self-training loop, and

also consider evidence integration regimes that turn off self-training (cf. Section Evidential learning).

We track the Popper metrics (Section Popper metrics), the consistency metrics (Section Consistency metrics), and the Bayesian metrics (Section Bayesian metrics) continuously during PHASE 2 and PHASE 3.

## Corpus construction

We construct synthetic pre-training corpora in language $L$ with a domain $D$ of constant size $k = 200$ by simulating authors who produce texts. The constructed corpora vary, as will be detailed below, in terms of inter-textual semantic agreement (the extent of which is controlled by the ratio of authors' background knowledge, $r_{bg}$) and the degree of inferential closure (controlled by the so-called `reach` threshold).

Constructing a corpus in our language $L$, we fix, first of all, a random permutation of $r_{bg} \cdot k$ items sampled from the domain $D$ as shared background knowledge $K$ of all authors ($n = 15$). We construct, likewise semantically, the belief state $B_i$ of author $i = 1 \dots n$ as a randomly chosen total strict order on a subset of $D$ which extends the order induced by $K$ (thus, allowing for belief suspension). The ratio of background knowledge, $r_{bg}$, hence controls the diversity of beliefs held by the simulated authors.

We say that a statement $\varphi$ is $i$-expressible iff $\varphi$ is believed by author $i$ and the *rank-order difference according to $B_i$* between the items referred to in $\varphi$ is smaller than the global `reach` threshold. The `reach` parameter constrains text generation and regulates, in particular, the degree of inferential closure of text corpora. With reach $< k$, a corpus is (in general) not deductively closed, i.e., there exist statements that are not contained in, but can be inferred from the corpus.

An author $i$ produces finite, truthful, unbiased, consistent, inferentially structured, nearly non-redundant $L$-texts, i.e., sequences of $i$-expressible $L$-sentences $\varphi_1, \varphi_2, \dots, \varphi_l$. Texts are truthful because they only contain sentences an author $i$ believes to be true ($B_i \vDash \varphi_j$ for $j = 1 \dots l$). Texts are unbiased because all statements which (i) the author can express given the `reach` threshold and which (ii) the author considers as true are equally likely to figure in a text by the author. Texts are consistent because whatever they contain is believed by the author, whose belief state is globally consistent. Texts are inferentially structured because, rather than expressing an author's beliefs in random order, texts follow the logical implications defined by $L$'s inference-rules, in particular, they contain transitivity arguments (e.g., $a_i R a_j$, $a_j R a_l$, $a_i R a_l$) and duality arguments ($a_i R a_j$, $a_j S a_i$) as sub-sequences. Texts are nearly non-redundant because, since they are unbiased, it is very unlikely that one and the same sentence figures twice in a text. S1 Algorithm. gives further details of how simulated authors sample texts.

All simulated authors ($n = 15$) contribute an equal share of texts to a corpus, which contains 100,000 training texts in total.

In the way described henceforth, we construct 60 corpora on a parameter grid with $r_{bg} \in \{0.05, 0.1, 0.15 \dots 0.75\}$ and reach $\in \{50, \infty\}$, and two corpora per parameter combination.

## Pre-training

To train RANKERS (basically T5 models) on the synthetic corpora, the raw texts have to be transformed into sequence-to-sequence data. We do so by defining a *denoising task* and a *text completion* task.

**Denoising task.**   In analogy to the pre-training task used by [1], we replace subsequences (1–2 tokens) of the raw text with special mask tokens. The masked text serves as input; the

target text details the correct substitution for each mask. For example:

$$\text{raw} : \quad a\ R\ b\ c\ S\ d\ a\ R\ e$$
$$\text{input} : \quad a\ [m1]\ c\ S\ d\ [m2]\ R\ e$$
$$\text{target} : \quad [m1]\ R\ b\ [m2]\ a\ [m3]$$

**Text completion task.** This task consists in completing and continuing a given text. The first $m$ sentences of a raw text (with $l$ sentences) serve as input (random $m$ with $0 < m < l$), the entire raw text is the target. For example:

$$\text{raw} : \quad a\ R\ b\ c\ S\ d\ a\ R\ e$$
$$\text{input} : \quad a\ R\ b\ c\ S\ d$$
$$\text{target} : \quad a\ R\ b\ c\ S\ d\ a\ R\ e$$

Every raw text in a corpus is transformed into one denoising example and one text completion example, yielding 200.000 pre-training items, on which a randomly initialized RANKER model is trained for 18 epochs, using the transformers framework [72]. S1 Fig displays eval loss during pre-training for all 60 models.

## Self-training

Self-training *pre evidence introduction*, i.e. PHASE 2, consists in 300 training steps and proceeds —with the exception of data augmentation—in close analogy to self-training in [61]. Self-training is designed so as to mimic a local belief revision procedure that consists in (i) identifying some strongly held beliefs, (ii) spelling out inferential consequences of these beliefs, and (iii) reinforcing one's beliefs in these consequences.

At each step in a self-training loop, the RANKER model generates texts, which are processed, filtered, masked, augmented and finally used as training data for denoising training (see S2 Algorithm). More specifically, we generate, first of all, 200 prompts by sampling strong beliefs from the RANKER (see also S3 Algorithm). Being queried with each of these prompts, the model returns, with beam sampling, 5 generated text sequences and corresponding scores. Texts are split into sub-sequences of length 3, discarding all sub-sequences which do not represent a syntactically well-formed sentence. Next, we keep only sentences from texts with at least 6 well-formed sentences and high beam scores (above 75th-percentile). These sentences are transformed into training data by masking their predicate letters—similarly to the masking for belief elicitation (cf. Section Conditional elicitation). These auto-generated training items are augmented with denoising examples from the pre-training corpus (ratio: 4 corpus sentences for 10 self-generated sentences). Finally, the RANKER model is trained with the thusly generated training items for one epoch.

## Evidential learning

In PHASE 3, RANKER models are exposed to *novel evidence*—which we represent through a single statement that contradicts formerly uncontested assumptions. Moreover, we distinguish different evidence integration regimes (EIR), i.e., ways in which RANKERS train on the novel evidence.

An **evidence item** $E$ to which a model is exposed in PHASE 3 is a single $L$-sentence. In our experimental set-up, it is chosen right at the beginning of PHASE 2 in such a way that it

contradicts the background knowledge $K$ underlying the corresponding, original pre-training corpus. More specifically, we randomly sample some $\varphi \in L$ such that $K \vDash \varphi$ and set $E := \bar{\varphi}$.

The statement $\varphi$ can be more or less strongly embedded, or "logically entrenched," in $K$: on the one extreme, $\varphi$ is logically independent from all the other sentences entailed by $K$; on the other extreme, many different sets of sentences entailed by $K$ in turn imply $\varphi$. In the second case, a revision of $\varphi$ has major logical repercussions for $K$ as a whole, whereas in the first case, $\varphi$ might simply be replaced with its negation in $K$ without running into any contradictions whatsoever. Now, we may exploit our simple semantics to define a straightforward measure of **evidential entrenchment** of an evidence item $E$ that negates some $\varphi$ (with $K \vDash \varphi$), namely the *rank order difference according to K* between the items referred to in $\varphi$.

So, to illustrate the concept of entrenchment, let $K$ be the strict order a b c d e f, then $K \vDash$ $aRb$, $bSa$, $bRc$, . . . (assuming $I(x) = x$ for every constant $x$). Sentence $cRd$, for example, having rank order difference 1, is minimally entrenched in $K$: replacing it with its negation ($dRc$) doesn't generate any inconsistencies. Sentence $cRf$, however, has rank order difference 3, it is more deeply entrenched and replacing it with its negation generates multiple inconsistencies, e.g., because $K \vDash cRd$, $dRf$.

Evidential entrenchment will turn out to be an important explanatory variable for understanding the models' responses to novel evidence (cf. Section Post evidence introduction (PHASE 3)).

We shall next describe the different ways in which RANKER models are exposed to evidence in PHASE 3, specifically, how an evidence item is integrated into the training data. The evidence integration regimes *prompt* and *append_gen* make use of the novel evidence item while *generating* texts for self-training:

**Prompt (EIR).** This regime continues self-training as carried out in PHASE 2, with the exception that a certain ratio ($r_{prompt}$ =.2) of prompts that are used to generate texts now contain—besides two further, strongly believed sentences—the evidence statement itself. Already in generating texts for self-training, models (may) respond to the novel evidence.

**Append_gen (EIR).** This regime, too, continues self-training as carried out in PHASE 2. However, the evidence item is appended to the pre-processed auto-generated texts and thereby integrated into the training data (denoising tasks).
Besides these two similar regimes, we consider—in what is sometimes called an ablation study—further EIRs that turn off self-training.

**Append_bel (EIR).** This regime appends the novel evidence item to texts (like *append_gen*), which are then used for further denoising training, except that, here, the training texts are not continously generated, but are simply sampled from (and thus reflect) the model's past beliefs *pre evidence introduction* (step = 300).

**Evidence_only (EIR).** Under this arguably most simple regime, the model merely trains on the evidence item itself.

## Results

### Pre evidence introduction (phase 2)

Self-training markedly improves the probabilistic coherence (cf. Section Probabilistic coherence below) and the logical consistency (cf. Section Logical alignment below) of RANKERS' degrees of belief.

**Probabilistic coherence.** We observe that self-training may substantially reduce initial violations of probabilistic coherence (as measured by Popper metrics). The epistemic benefits

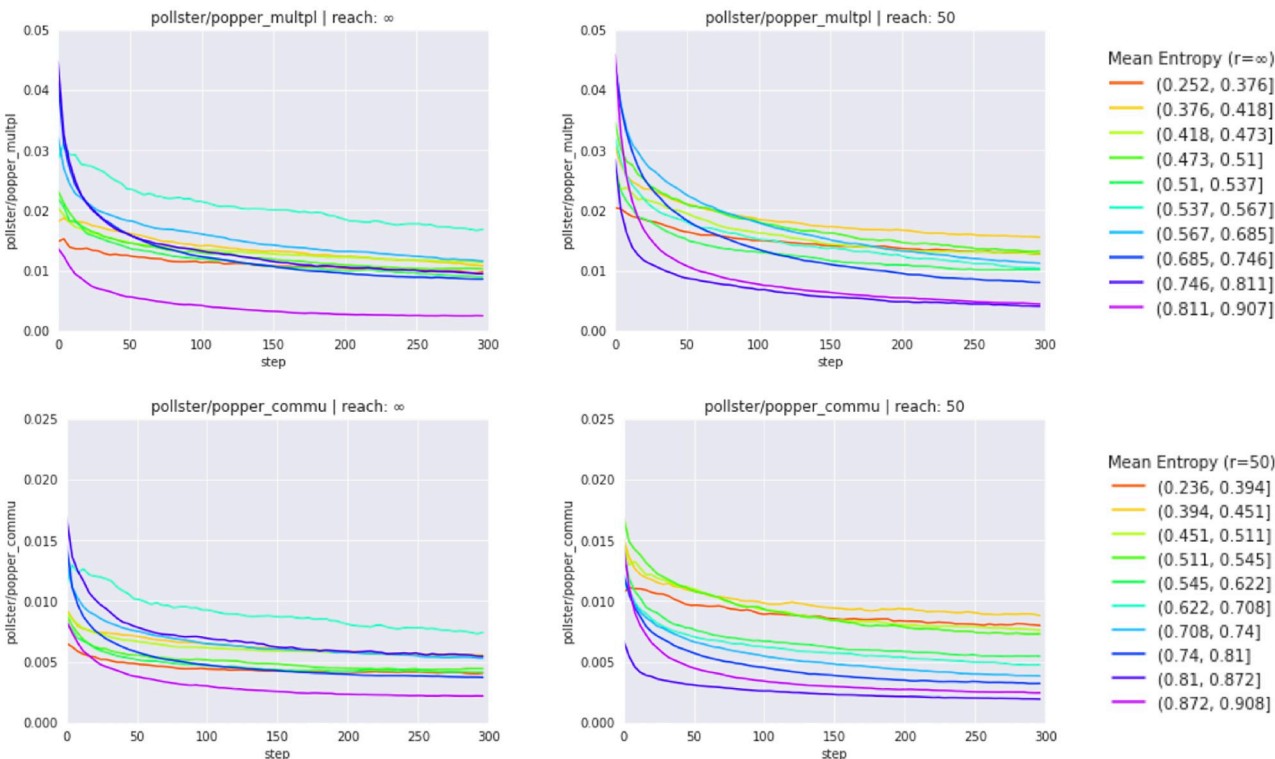

**Fig 2. Evolution of Popper metrics.** Evolution of Popper metrics POPPERCOMMU (*commutativity*) and POPPERMLTPL (*multiplication*) during self-training. Left: inferentially closed pre-training corpora (`reach=∞`); right: inferentially incomplete pre-training corpora (`reach=50`). Metric evolutions are aggregated over all agents whose belief systems display a similar doxastic entropy (cf. bins to the very right).

of self-training are especially transparent with respect to *multiplication* and *commutativity constraints* (Fig 2), which are poorly satisfied by pre-trained models.

Yet, even pre-trained RANKERS comply with *complementarity* (credences assigned to contradictory sentences add up to 1); self-training has no further effect here (S2 Fig). And all models, self-trained or not, struggle with *reflexivity* (S2 Fig)—though we should bear in mind that this may be an artifact of the synthetic training texts, which, being both consistent and *nearly nonredundant* (cf. Section Corpus construction), don't contain, by construction, any subsequence corresponding to a masked reflexivity-query (e.g., a R b a R b or a R b b S a).

**Logical alignment.** Self-training improves logical alignment of credences. It brings down, consistent with previous findings [61], the frequency of transitivity violations (Fig 3) and improves further consistency metrics (see also S3 Fig).

We find nevertheless that credences are poorly aligned with entailment relations (LogAlg-nEnt), and self-training does seemingly not improve these deficits, at least as judged by aggregate measures (Fig 3). This seems like an important flaw: If $P$ entails $\varphi$, then the conditional degree of belief in $\varphi$ given $P$ *should* be, but in fact is not close to 1.

However, a further analysis of pre-trained models reveals that $Bel(\varphi|P)$, with $P \vDash \varphi$, deviates from 1 especially in cases where sentence $\varphi$ is—unconditionally—strongly *dis*believed, i.e. if $Bel(\varphi) \approx 0$ (see S5 and S6 Figs). If, in contrast, $Bel(\varphi) \gg 0$, then $Bel(\varphi|P) \approx 1$! Plus, the belief in some statement is nearly always increased by conditionalizing on an antecedent that entails the very statement (see S7 and S8 Figs). This fine-grained analysis shows that degrees of belief are in fact well aligned with entailment relations, especially so as the *entailment* constraint (cf.

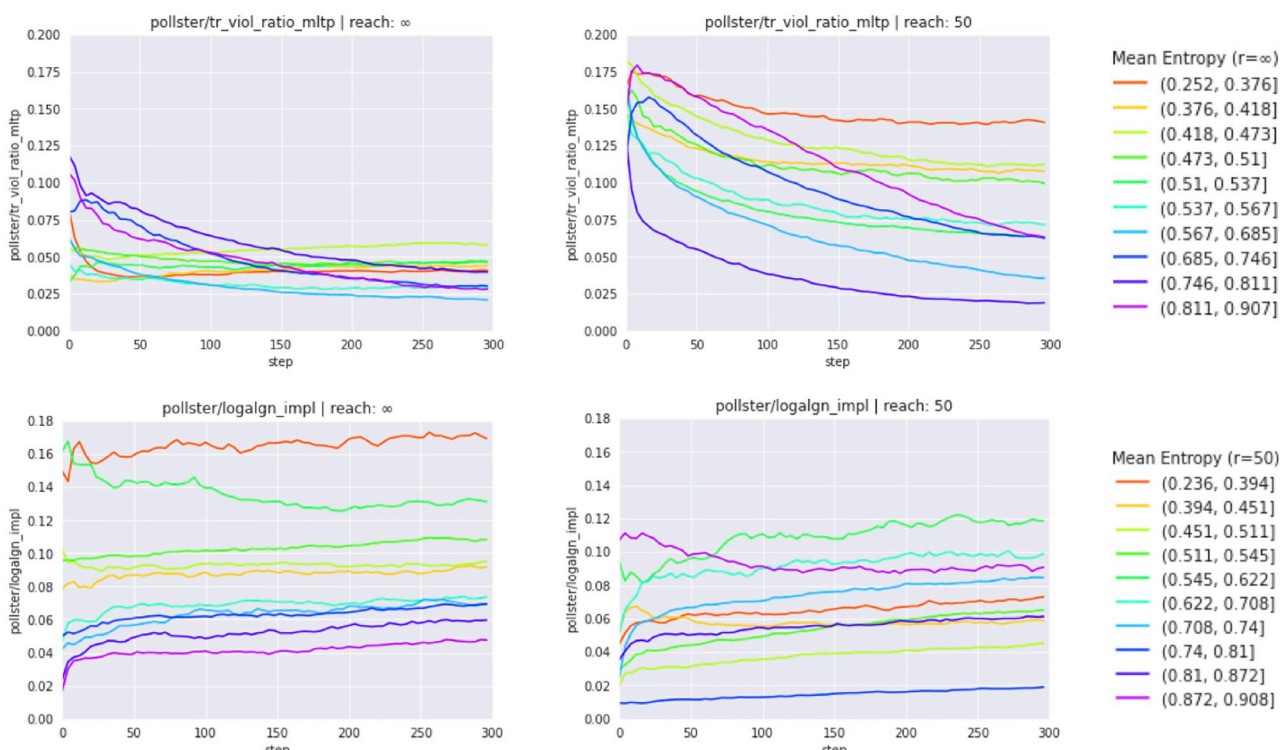

**Fig 3. Evolution of consistency metrics.** Evolution of consistency metrics TRANSVIOLRATIO (*transitivity violation ratio*) and LOGALGNENT (*entailment alignment*) during self-training. Left: inferentially closed pre-training corpora (`reach=∞`); right: inferentially incomplete pre-training corpora (`reach=50`). Metric evolutions are aggregated over all agents whose belief systems display a similar doxastic entropy (cf. bins to the very right).

Section Consistency metrics) applies to regular, non-extreme unconditional credences only ($Bel(\varphi|P) \neq 0$).

## Post evidence introduction (phase 3)

Integrating evidence items into self-training loops allows RANKERS to acquire novel evidence that contradicts previously held beliefs *without* sacrificing probabilistic coherence (Section Probabilistic coherence below) and logical consistency (Section Logical alignment below), as well as to propagate this novel evidence through their belief systems in *approximate* agreement with Bayesian constraints (Section Evidential learning below). The following sections report findings for the evidence introduction regime *prompt*; the regime *append_gen*, which relies equally on self-training and only slightly modifies *prompt*'s training data generation, yields highly similar results that are presented in the Appendix and will be referred to where appropriate. The evidence introduction regimes *append_bel* and *evidence_only*, which turn off self-training, represent ablation studies and are discussed separately (Section Ablation studies).

**Probabilistic coherence.**   Comparing the periods *pre* (PHASE 2) and *post* (PHASE 3) evidence introduction, we find no statistically significant deterioration in the Popper metrics (see S1 Table). Under the evidence introduction regime *prompt*, a model's response to novel evidence doesn't decrease probabilistic coherence (see also S11 Fig).

**Logical alignment.**   Comparing the periods *pre* (PHASE 2) and *post* (PHASE 3) evidence introduction, we do find, regarding consistency metrics, a statistically significant deterioration (see

) only for metric LᴏɢAʟɢɴEɴᴛ (which systematically overestimates the model's flaws, see Section Logical alignment above), and for metric TʀᴀɴsVɪᴏʟRᴀᴛɪᴏ (`reach=∞`). Yet the latter increase in the *frequency* of transitivity violations is partially compensated by a simultaneous reduction in the *average extent* of transitivity violations as measured by TʀᴀɴsVɪᴏʟMSE (which had already been brought down to a low level in ᴘʜᴀsᴇ 2, before). All in all, under the evidence introduction regime *prompt*, a model's response to novel evidence doesn't substantially deteriorate logical consistency (see also ).

**Evidential learning.**   To study evidential learning, we track the evolution of four metrics *pre* and *post evidence introduction*: the three Bayesian metrics (introduced in Section Bayesian metrics), and the model's absolute degree of belief in the evidence statement itself.

We focus, in analyzing the propagation of evidence through the belief system, on *statements $\varphi$* such that the model's prior absolute belief in $\varphi$ deviates substantially from the model's prior conditional belief in $\varphi$ given the evidence $E$—or, in other words, on statements with high KLsyn-average *pre evidence introduction* (see Section Bayesian metrics). It's with respect to these cases that we can study most clearly how the model adjusts its beliefs, and whether it adheres to diachronic constraints of Bayesian learning. In the subplots of , each of the three lines (solid, dashed, dotted) focuses on a corresponding top $q$th-percentile of pre-evidence KLsyn: the greater $q$, the more absolute and conditional belief initially diverge.

Moreover, we distinguish models according to the inferential closure of their pre-training corpus (`reach`), the evidence introduction regime deployed in ᴘʜᴀsᴇ 3, and the evidential entrenchment, i.e., the degree of entrenchment of the background assumption which is contradicted by the new evidence statement eventually introduced (see Section Evidential learning).

 plots the evolution of Bayesian metrics with weak evidential entrenchment (left-hand side, below 25th-percentile entrenchment) versus strong evidential entrenchment (right-hand side, above 75th-percentile entrenchment). Qualitatively speaking, the metrics evolve in similar ways: First of all, as shown in the bottom plots (red lines), the degree of belief in the

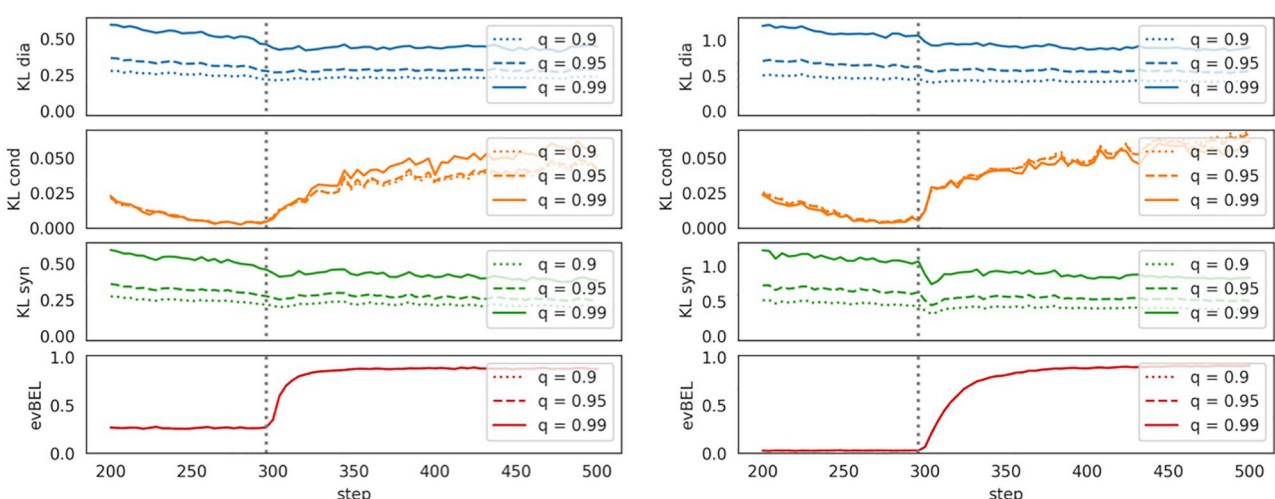

**Fig 4. Evolution of Bayesian metrics.** Evolution of Bayesian metrics before and after evidence introduction with evidence introduction regime *prompt* and inferentially closed pre-training corpora (`reach=∞`). Left: evidence contradicts a weakly entrenched background assumption; right: evidence contradicts a strongly entrenched background assumption.

evidence *E* starts to increase as soon as the evidence is introduced, and the models eventually come to fully believe *E*. Secondly, the KLsyn trajectories (green lines) drop upon evidence introduction, which signifies that absolute beliefs and conditional beliefs (given the evidence) are gradually aligned to each other. However, and thirdly, this synchronous mutual alignment (measured by KLsyn) is at least partially due to changes in conditional beliefs (KLcond, orange lines): Conditional beliefs start to deviate from their pre-evidence level after evidence introduction. Fourthly, the KLdia trajectores (blue lines) decrease clearly after evidence introduction: the models tend to align their absolute beliefs with the *prior* conditional beliefs as prescribed by Bayesian learning. Yet, diachronic alignment (KLdia) seems to be somewhat less pronounced than synchronic alignment (KLsyn).

So the general upshot so far (further confirmed by S9 Fig) is that the models fully adopt the evidence item they are supposed to learn, and that they adjust their other beliefs so as to synchronously align absolute and conditional beliefs (given the novel evidence). This synchronous convergence is achieved both through changes in conditional beliefs (in violation of simple conditionalization) and by aligning absolute beliefs with corresponding priors (proper Bayesian learning). This said, the models fall short of being perfect Bayesian agents (which would require that metrics approach 0 after evidence introduction)—even the weak synchronous constraint on evidential learning is substantially and continuously violated: they are, at most, noisy Bayesian agents.

Can we also make sense of the differences we observe when comparing the left-hand and the right-hand panels of Fig 4? We should first of all note that the two groups of self-training / evidential learning runs (corresponding to the two panels) differ not only with respect to evidential entrenchment, but also with regard to the average PHASE 2-entropy of the model's global belief system (*H*)—which stems from a selection bias: deep entrenchment is only possible if the pre-training corpus contains a sufficiently broad background knowledge, which in turn results in strong beliefs and low global doxastic entropy. This being said, the following differences *pre evidence introduction* seem coherent: evidence items, even though contradicting background assumptions, are much more strongly believed in case of weak entrenchment and high entropy (compare the initial levels of evBel); similarly, the higher doxastic entropy, that is, the less degrees of belief tend to differ from.5, the smaller the Bayesian divergence metrics (cf. initial levels of KLsyn and KLdia). Regarding *post evidence introduction*, it seems consistent that models much more quickly adopt an evidence item they had *partially* believed rather than an evidence item they had fully *dis*believed (see the steep rise in evBEL trajectories for low entrenchment). Moreover, the revision of a poorly entrenched belief should trigger—for lack of inferential relations—fewer and less pronounced revisions of other beliefs, which is precisely what we observe when comparing the pace and extent of changes in the Bayesian metrics after evidence introduction.

In contrast to the highly aggregated analysis presented henceforth, S10 Fig plots *individual* (rather than mean) KLsyn trajectories for each statement (N = 1024) in a sample during PHASE 2 and PHASE 3, focusing on 16 self-training runs with four different models (each trained in a corpus with $r_{bg}$ = 0.5). Even for one and the same model and for evidence items with comparable entrenchment, we may observe strikingly different patterns of belief adjustment, e.g. regarding the durability of an immediate decrease in KLsyn after evidence introduction. This clearly points to the limits of our current analysis. It is, for instance, not clear whether these different dynamical patterns stem from properties of the auto-generated texts used for training, from the way the evidential statement is embedded in the model's actual belief system (rather then in the background theory underlying the pre-training corpus), or whether the different patterns represent a measurement artifact caused by small sample sizes.

## Ablation studies

Our ablation studies, which vary the evidence introduction regime, confirm the pivotal role of self-training for the consistent integration of novel evidence into a RANKER's belief system: Evidential learning seems not feasible without self-training.

The regime *append_gen* retains the self-training loop and merely modifies how an evidence item is merged into the training data. Rather than integrating the evidence right in the prompts used for generating texts (as with regime *prompt*), *append_gen* simply appends the evidence statement to generated texts. This has only a minor effect: The Popper and consistency metrics are as good as (if not slightly better than) with *prompt* (S11 and S12 Figs), and the model partially succeeds in Bayesian evidential learning (S9 Fig).

However, things look entirely different as soon as we turn off self-training. With the regime *append_bel*, the novel evidence is not appended to an auto-generated text, but to varying sets of beliefs of the model elicited right before evidence introduction: The RANKER model trains on its previous beliefs and the new evidence. This leads to a totally uninformative and inconsistent belief state. The doxastic entropy of the model is maximal (S13 Fig), while consistency metrics deteriorate (S12 Fig). Transitivity metrics, in particular, more than quadruple.

The second regime which does without self-training is *evidence_only*: RANKERS simply train on the evidence statement. Here, the model fails to consistently integrate the evidence as well: Both the Popper metrics and the consistency metrics increase—i.e., worsen—dramatically (S11 and S12 Figs), leading to a probabilistically incoherent and logically inconsistent belief state.

## Discussion

The computational experiments with NLMs conducted in this paper provide evidence for the *Rationality Hypothesis*, i.e., the claim that suitably trained neural language models are epistemic agents, exhibiting advanced rational behavior. For these purposes, we have pre-trained T5 text-to-text models—so-called RANKERS—from scratch on carefully designed synthetic corpora composed of internally consistent texts (PHASE 1). Subsequently, the pre-trained RANKERS have been submitted to self-training on auto-generated texts (PHASE 2). Finally, we have exposed the self-trained RANKERS to novel evidence (PHASE 3), choosing evidence items which contradict the uncontested statements contained in the original pre-training corpus, and which are hence strongly disbelieved by the self-trained models. Our main concern of interest has been whether the models, during pre- or self-training, gradually acquire probabilistically coherent and logically consistent credences, and whether these degrees of belief, once consolidated, are rationally revised given novel (unexpected) evidence. The latter challenge does not just amount to adapting the degree of belief about the evidence statement itself, but consists, moreover, in simultaneously re-adjusting the entire belief system so as to avoid probabilistic incoherence and logical inconsistency.

We may summarize our results as follows.

**Q1.** Are the degrees of belief elicited from a pre-trained RANKER both probabilistically coherent and logically consistent (i.e., aligned with the language's logico-semantic constraints), provided that the RANKER itself has been trained on consistent texts?

No. The pre-trained models fail to aggregate collectively diverse, internally consistent texts into a coherent and consistent probabilistic belief system. We observe substantial violations of probabilistic constraints (e.g., regarding the multiplication rule and commutativity) and widespread inconsistency (more specifically, and depending on the precise model, more than 15% of the transitivity constraints may be violated). These results agree with previous findings [61],

which suggest that NLMs' inconsistent judgments emerge because pre-training emulates sentence-wise judgment aggregation.

**Q2.** Does self-training improve probabilistic coherence and logical consistency of degrees of belief?

Yes. We see clear progress regarding most Popper and consistency metrics during self-training. And in case self-training doesn't improve a specific metric (such as logical alignment with entailment relation, reflexivity, or complementarity), we can explain the lack of improvement, namely with reference to low initial levels, or by means of data segmentation and in-depth analysis. The observed logical benefits of self-training are consistent with [61, 73].

**Q3.** Can RANKERS successfully integrate novel evidence into their belief system without loosing probabilistic coherence and logical consistency?

Yes. To answer this question, we have explored alternative evidence integration regimes. If the novel evidence is integrated into the self-training loop, and the model effectively trains on the evidence item plus continuously auto-generated texts, RANKERS succeed well in (a) adopting novel evidence, and (b) propagating novel evidence through the entire belief system so as to preserve consistency and coherence. If, however, RANKERS train on novel evidence without simultaneous self-training, the models newly adopt the evidence, but run into a highly inconsistent as well as incoherent belief state, while risking, in addition, to loose any informational content (maximal entropy). These experiments underline the pivotal role of self-training for rational, coherence-preserving evidential learning.

**Q4.** Do RANKERS adjust their belief state given novel evidence in accordance with Bayesian learning (conditionalization)?

Yes, to some extent. With self-training turned on in PHASE 3, models *tend* to pay attention to synchronic and diachronic constraints of evidential learning. Signs of Bayesian evidential learning are most pronounced immediately after evidence introduction, and in cases where the prior belief that is directly negated by the novel evidence is deeply entrenched in the background knowledge of the original pre-training corpus. A more detailed analysis of individual ensemble runs reveals, however, stark differences between single runs with similar boundary conditions, suggesting that RANKERS' response to novel evidence might be further governed by variables not controlled for in our experiments (such as, e.g., the entrenchment of an evidence item in a model's actual belief system, or further inferential properties of the randomly auto-generated training texts).

We may conclude: Qua self-training, models carry out piecemeal adjustments of their credences so as to iron out local inconsistencies, gradually improving the global coherence of their belief state. It has been argued [61] that such self-training can be understood as a simple form of reflective equilibrium (cf. [74, 75]). Continuous reflective equilibrium turns our simple RANKERS into epistemic agents that, having acquired a language and proto-beliefs, form a coherent and consistent belief system, and rationally revise it given novel evidence.

While the experiments reported in this paper confirm the *Rationality Hypothesis*, the evidence they provide is limited due to the strong idealizations they rely on, a lack of systematic robustness analysis, and apparent explanatory gaps in understanding the detailed simulation results:

- We simulate (arguably complex) probabilistic belief dynamics by training T5 models on a **most simple language** with minimal syntax and logic. Our experiments don't settle the

question whether NLMs which are suitably trained on more complex languages, are epistemic agents, too.

- Moreover, regarding **robustness**, the ensemble of models we study doesn't establish whether our findings hold for larger NLMs, or for transformers with a different configuration (e.g., BART, GPT-3). We only cover corpora with particular agreement patterns, and haven't systematically varied the auto-generation of texts (e.g., decoding parameters), either.

- This study lacks the resources to analyze and explain the **micro-dynamics** of belief formation and, specifically, revision. As one and the same model may display starkly different responses to novel evidence items (given the same EIR), the micro-dynamics, remaining by and large opaque, may however determine the degree of rationality of a model's particular response to novel evidence.

- We train RANKERS not only on a simple, but also on an **artificial language**. Properties of natural languages (such as, e.g., synonymity, homonymy, intensional contexts) might prevent NLMs trained on, say, English text corpora from acquiring advanced rational faculties.

All this puts our experiments in perspective: In more complex and more realistic, or micro-dynamically different experimental settings (than those studied here), NLMs might fail to display advanced rational behavior. While this would not amount to a refutation of the *Rationality Hypothesis*, it would severely limit its scope and hence its conceptual, normative, and empirical fruitfulness.

All this calls for further exploration. We shall close by outlining five directions of future research:

1. Follow-up investigations may explore the robustness of the synthetic experimental set-up, especially by reproducing the experiments with a richer and more expressive artificial language (containing, e.g., FOL, epistemic operators, a truth predicate, or indirect speech).

2. Similarly, it seems important to further investigate the epistemic benefits of self-training in natural languages. To begin with, NLMs might be trained from scratch on a clearly delineated fragment of the English language (e.g., causal reasoning in a limited physical domain), which will probably involve the usage of synthetic texts. But probabilistic belief formation and revision should also be studied in NLMs that have been pre-trained on large natural language corpora: In that case, belief elicitation might be initially restricted to domains corresponding to well-defined NLP-tasks (such as NLI), where extensive datasets are available. The fact that machine translation benefits from iterative back-translation [73] and training on pseudo-labels may improve predictive performance [76, 77] is further evidence for the fruitfulness of self-training as a learning paradigm.

3. Improved diagnostic tools and further experiments are called for to better understand the micro-dynamics of NLMs' belief revisions. Such diagnostic tools would include, e.g., high-resolution representations of an agent's probabilistic belief system, and should allow one to relate the semantic content and the inferential structure of training texts to the current belief state of a model. Moreover, further ablation studies might replace auto-generation (in order to better understand its effect) with rule-based, experimentally controlled generation of training data.

4. Because of its mathematical affinity to probabilistic word prediction machines, Bayesian epistemology suggests itself as a normative framework for testing the rationality of NLMs. However, it seems worthwhile to assess NLMs with respect to further standards of epistemic agency: Do NLMs, in revising their beliefs, opt for the best available explanation [78]? Do

NLMs acquire beliefs that mutually support each other, rather than merely satisfying probabilistic and logical constraints [79, 80]? Can NLMs' credences be accounted for in terms of ranking theory [81]? When faced with rival hypotheses, do NLMs acquire beliefs in accordance with principles of rational scientific theory choice [82]? Can NLMs effectively engage in causal reasoning [83]? Do NLMs succeed in acquiring accurate credences that are well-calibrated relative to a ground-truth [84, 85]? May NLMs exhibit socio-epistemic rationality, such as diagnosing and responding to peer disagreement, or resolving conflicting testimony by means of competence ascriptions [86]?

5. Finally, a farther step beyond this study would be to let multiple self-training NLMs, which form and revise their credences in parallel, interact with each other: reading and grading each others' texts, generating written responses, engaging in dialogues, and training on an individually selected subset of the collectively and continuously generated text data. Such a set-up might allow one to study the contribution of social interaction to the emergence of rationality, extending the work on the emerhence of language in deep multi-agent systems [87].

## Supporting information

**S1 Appendix. Credences of non-atomic sentences.** Description of how degrees of belief can be assigned to non-atomic *L*-sequences.
(PDF)

**S2 Appendix. Measure of probabilistic incoherence.** Discusses Popper metrics as proxies for global probabilistic incoherence.
(PDF)

**S3 Appendix. Transitivity constraint.** Motivation of the transitivity constraint for credences.
(PDF)

**S1 Algorithm. Text-production by simulated author.**
(PDF)

**S2 Algorithm. Self-training loop.**
(PDF)

**S3 Algorithm. Prompt construction.**
(PDF)

**S1 Fig. Eval loss during pre-training.** Eval loss during pre-training, each line corresponds to one of 60 models.
(PDF)

**S2 Fig. Evolution of Popper metrics *complementarity* and *reflexivity* during self-training.**
Left: inferentially closed pre-training corpora (reach=∞); right: inferentially incomplete pre-training corpora (reach = 50). Metric evolutions are aggregated over all agents whose belief systems display a similar joint entropy (cf. bins to the very right).
(PDF)

**S3 Fig. Evolution of Consistency Metrics *transitivity violation ratio,equivalence*, and *antecendent equivalence* during self-training.** Left: inferentially closed pre-training corpora (reach=∞); right: inferentially incomplete pre-training corpora (reach = 50). Metric evolutions are aggregated over all agents whose belief systems display a similar joint entropy (cf.

legends in S2 Fig).
(PDF)

**S4 Fig. Global dynamics (evolution of joint entropy, global distance, volatility) during self-training.** Left: inferentially closed pre-training corpora (reach=∞); right: inferentially incomplete pre-training corpora (reach = 50). Metric evolutions are aggregated over all agents whose belief systems display a similar joint entropy (cf. legends in S2 Fig).
(PDF)

**S5 Fig. Logical alignment of pretrained models.** Logical alignment of pretrained models (step = 0) as a function of reach (rows) and number of distractors (columns). For each model, 1000 inferences (premises, conclusion, distractors) are sampled, and for each inference, the unconditional degree of belief in the conclusion (x-axis) as well as the conditional degree of belief given premises and distractors (y-axis) are elicited and shown.
(PDF)

**S6 Fig. Logical alignment of pretrained models.** Logical alignment of pretrained models (step = 0) as a function of background knowledge ratio in the training corpus (rows) and number of distractors (columns). For each model, 1000 inferences (premises, conclusion, distractors) are sampled, and for each inference, the unconditional degree of belief in the conclusion (x-axis) as well as the conditional degree of belief given premises and distractors (y-axis) are elicited and shown.
(PDF)

**S7 Fig. Logical alignment of pretrained models.** Logical alignment of pretrained models (step = 0) as a function of reach (rows) and number of distractors (columns). For each model, 1000 inferences (premises, conclusion, distractors) are sampled, and for each inference, the unconditional degree of belief in the conclusion (x-axis) as well as the conditional degree of belief given premises and distractors are elicited. The y-axis show the absolute difference between conditional and unconditional beliefs.
(PDF)

**S8 Fig. Logical alignment of pretrained models.** Logical alignment of pretrained models (step = 0) as a function of background knowledge ratio in the training corpus (rows) and number of distractors (columns). For each model, 1000 inferences (premises, conclusion, distractors) are sampled, and for each inference, the unconditional degree of belief in the conclusion (x-axis) as well as the conditional degree of belief given premises and distractors are elicited. The y-axis show the absolute difference between conditional and unconditional beliefs.
(PDF)

**S9 Fig. Evolution of Bayesian metrics before and after evidence introduction.** Evidence introduction regimes: "prompt" (rows 1,2); and "append_gen" (rows 3,4). Inferential closure of pre-training corpora: reach=∞ (row 1,3); reach = 50 (rows 2,4). Evidential entrenchment: columns correspond to quartiles.
(PDF)

**S10 Fig. Synchronous KL divergence before and after evidence introduction.** Synchronous KL divergence (between unconditional belief and conditional belief given the evidence) before and after evidence introduction (evidence introduction regime "prompt") for 1024 individual beliefs elicited in four selected models (pre-trained on corpora with background ratio 0.5). Each row displays four different self-training runs of one and the same model.
(PDF)

**S11 Fig. Comparison of Popper metrics before and after evidence introduction.** Comparison of Popper Metrics before and after evidence introduction for different evidence introduction regimes. Left column: inferentially closed pre-training corpora (reach=∞); right column: inferentially incomplete pre-training corpora (reach = 50).
(PDF)

**S12 Fig. Comparison of Consistency metrics before and after evidence introduction.** Comparison of Consistency Metrics before and after evidence introduction for different evidence introduction regimes. Left column: inferentially closed pre-training corpora (reach=∞); right column: inferentially incomplete pre-training corpora (reach = 50).
(PDF)

**S13 Fig. Comparison of joint entropy and evidence belief before and after evidence introduction.** Comparison of joint entropy and evidence belief before and after evidence introduction for different evidence introduction regimes. Left column: inferentially closed pre-training corpora (reach=∞); right column: inferentially incomplete pre-training corpora (reach = 50).
(PDF)

**S1 Table. Statistical comparison of metrics pre and post evidence introduction.** Statistical comparison of metrics pre (steps 0–299) and post (steps 300–600) evidence introduction for evidence integration regime "prompt": standard deviation of metric pre evidence introduction (STD PRE); difference in means post vs pre (DELTA MEAN); p-value of t-test for independence of pre and post metric samples (p); mean difference relative to standard deviation (DELTA/STD); boolean variable indicating statistically significant ($p < 0.05$) deterioration of metric (sign).
(PDF)

## Acknowledgments

We would like to thank the participants of the research seminar at KIT's Department of Philosophy, and especially Christian Seidel, for their astute comments regarding and earlier version of this manuscript.

Moreover, the thorough and constructive feedback of two reviewers helped us to improve the paper.

## Author Contributions

**Conceptualization:** Gregor Betz, Kyle Richardson.

**Data curation:** Gregor Betz.

**Formal analysis:** Gregor Betz, Kyle Richardson.

**Investigation:** Gregor Betz, Kyle Richardson.

**Methodology:** Gregor Betz, Kyle Richardson.

**Supervision:** Kyle Richardson.

**Writing – original draft:** Gregor Betz.

**Writing – review & editing:** Gregor Betz, Kyle Richardson.

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
