## [Decision Letter · Decision Letter 0]

29 Sep 2022

PONE-D-22-15831Probabilistic Coherence, Logical Consistency, and Bayesian Learning: Neural Language Models as Epistemic AgentsPLOS ONE

Dear Dr. Gregor Betz,

Thank you for submitting your manuscript to PLOS ONE. After careful consideration, we feel that it has merit but does not fully meet PLOS ONE’s publication criteria as it currently stands. Therefore, we invite you to submit a revised version of the manuscript that addresses the points raised during the review process.

We look forward to receiving your revised manuscript.

Kind regards,

Dr. Anu Sayal, Ph.D.

Academic Editor

PLOS ONE

Journal Requirements:

This work is supported by the Helmholtz Association Initiative and Networking Fund on the

HAICORE@KIT partition.

However, funding information should not appear in the Acknowledgments section or other areas of your manuscript. We will only publish funding information present in the Funding Statement section of the online submission form. 

This work is supported by the Helmholtz Association Initiative and Networking Fund on the HAICORE@KIT partition. The funders had no role in study design, data collection and analysis, decision to publish, or preparation of the manuscript.

NO

7. Please amend your list of authors on the manuscript to ensure that each author is linked to an affiliation. Authors’ affiliations should reflect the institution where the work was done (if authors moved subsequently, you can also list the new affiliation stating “current affiliation:….” as necessary).

8. Please ensure that you refer to Figure 1,2,3,4,5,6,7,8,13,14,15,16 and 17 in your text as, if accepted, production will need this reference to link the reader to the figure.

Reviewers' comments:

Reviewer's Responses to Questions

**Comments to the Author**

1. Is the manuscript technically sound, and do the data support the conclusions?

Reviewer #1: Yes

Reviewer #2: Yes

2. Has the statistical analysis been performed appropriately and rigorously? 

Reviewer #1: Yes

Reviewer #2: Yes

3. Have the authors made all data underlying the findings in their manuscript fully available?

Reviewer #1: Yes

Reviewer #2: Yes

4. Is the manuscript presented in an intelligible fashion and written in standard English?

Reviewer #1: Yes

Reviewer #2: Yes

5. Review Comments to the Author

Reviewer #1: This is an impressive and interesting article, making (as far as I know) novel both conceptual and technical points about the way in which the probabilistic and Bayesian coherence of NLMs may be assessed. I suspect it warrants publication, but I have a few conceptual questions that it may help for the authors to clarify before publication.

1) What exactly is the force of the Rationality Hypothesis? As the authors say in defense of it, it is an existential hypothesis about what a possible NLM might compute, in the right circumstances. But neural nets are Turing complete, meaning of course there will be SOME net that computes any function we like, including a function that outputs probabilistically coherent credences and adjusts them in a Bayesian fashion. So what exactly is the form of the hypothesis that is uncertain? Is the idea that such a neural net may require too different an architecture (or training regimen) from standard NLMs to count as an NLM? I think it'd help understand the point of the paper better to have a bit more discussion on this front—though obviously there are difficult issues here and I don't mean to hold the publication of the paper hostage to settling all of them perfectly clearly!

Perhaps here's another way to put the conceptual worry. The authors motivate their proposal by suggesting that it bears on whether neural nets may eventually exhibit AGI, since AGI agents must be rational. But it's trivially easy to construct programs that meet all our author's metrics perfectly: simply define a (non-neural-net) program that outputs a probability function over state desriptions in a Boolean langauge, and calculates probabilities of arbirary sentences by sums of their state descriptions. (Of course, this is computationally HARD, since the number of state descriptions explodes with the number of atomic sentences; but it's clearly programmable.) The fact that we can define such a Bayesian program doesn't seem to provide much if any evidence that it's possible to generate an AGI using a similar architecture. So why think the neural net case is any different?

2) Somewhat relatedly, I was surprised that the language did not contain Boolean operations. After all, that's what I'd initially expect was needed for a neural net to be "learning to be logically coherent". Why was this choice made? Is there a principled difficulty with running similar analyses with a more complex language, or is it simply an implementational difficulty? A bit more discussion would be helpful.

3) Why exactly does the belief elicitation protocol make sense, wherein we get the network's credences as it's probabilistic prediction for the completion of a masked string? Imagine you had a HUMAN doing this task: reading a bunch of sentences, and then predicting the next sentence. Clearly their prediction wouldn't necessarily line up with their degree of belief that the sentence is true; rather it's something closer to their degree of belief that "in this context, this corpus will present me with this string', or something like that. Is the thought that if they fully trust the corpus, that would be equivalent to their credence? I'm not sure that's right either. Difficult issues, we won't get to the bottom of them here. But more discussion for this way of measuring, and what it presupposes, would be nice.

4) The measures of deviation from probabilistic coherence are interesting, and touch on a not-huge-but-growing literature on what the proper way to define such measures is. It'd be nice to have some comparisons of the authors metrics to other ones—I suspect they will have important theoretical differences, but perhaps will not deviate too much in practice. (Though it'd be interesting if they did!) Some places to look for literature on this:

- Staffel 2015, "Measuring the overall incoherence of credence functions"

- Staffel and de Bona, "Graded Incoherence for Accuracy-Firsters'

5) I don't understand the transitivity constraint on page 18. It seems to be saying that if {A,B} entails C, then Pr(C) must be at least as great as Pr(A)*Pr(C). Right? But that is not a constraint of probabilistic coherence, and is violated whenever A and B are negatively relevant. Eg suppose we have a coin that's either biased 90% for H or biased 90% for T, and we're 50-50 on which it is. Then Pr(Heads first toss) = 0.5 and Pr(Tails second toss) = 0.5, but Pr(Heads first, Tails second) = 0.09 (if H-biased, it's 0.9*0.1; if T-biased, it's 0.1*0.9; the average of those is 0.09).

The appendix mentioned something about independence, but why should that be assumed in this context?

6) The discussion of the presentation of new evidence went by a bit quick for me (but perhaps I was reading quickly/less-carefully as that point, so don't put too much weight on this). Intuitively, the networks are getting new evidence at the earlier stages too—namely, when they're learning what the beliefs of the informant generating the sentences are. So conceptually I found it strange that presenting new evidence later is any different than the earlier stages. Is it just because at the earlier stage we're letting the network piggyback on the informant's logical coherence, and later we want to see if they can maintain coherence on their own? Perhaps just a word or two more on this would help tired readers understand what's happening.

Anyways, thanks for the thought-provoking paper!

Reviewer #2: This paper examines how well certain types of natural language models can acquire characteristics of bayesian (epistemic) rationality. The paper uses RANKERS and a simple artificial language to test properties such as general probabilistic coherence and updating by conditionalization.

The results are intriguing enough for me to recommend publication, but I overall was not fond of how the paper was written. The opening few pages state broad and grand theses that are ill-defined. Indeed, the Rationality Hypothesis---nominally the main subject of the paper---is stated in vague and unclear language and not made at all clear until 4 pages later (p. 6 of the ms). Some connections the authors draw were also very underspecified. For instance, on p. 3 line 63, the authors discuss Systems 1 and 2 in psychology, and it was entirely opaque to me how this investigation related to these concepts. So, I'd like to the authors being a little less grand and a little more precise in the opening pages.

I have some reservations as well about the conclusions. For example, the authors use an extremely limited artificial language and then present the Ranker with fully coherent text and probe its partial beliefs. It isn't especially surprising if a system with such an artificial language that is presented a variety of opposing, but fully consistent passages ends up in the convex hull of the views it's presented with. While the results are interesting, I think the significance is a bit limited.

6. PLOS authors have the option to publish the peer review history of their article (what does this mean?). If published, this will include your full peer review and any attached files.

Reviewer #1: **Yes: **Kevin Dorst

Reviewer #2: No

---

## [Author Response · Author response to Decision Letter 0]

3 Dec 2022

We've replied to the excellent comments by the reviewers in the separately provided response.

---

## [Decision Letter · Decision Letter 1]

23 Jan 2023

Probabilistic Coherence, Logical Consistency, and Bayesian Learning: Neural Language Models as Epistemic Agents

PONE-D-22-15831R1

Dear Dr. Gregor Betz,

We’re pleased to inform you that your manuscript has been judged scientifically suitable for publication and will be formally accepted for publication once it meets all outstanding technical requirements.

Kind regards,

Anu Sayal, Ph.D.

Academic Editor

PLOS ONE

Additional Editor Comments (optional):

Reviewers' comments:

Reviewer's Responses to Questions

**Comments to the Author**

1. If the authors have adequately addressed your comments raised in a previous round of review and you feel that this manuscript is now acceptable for publication, you may indicate that here to bypass the “Comments to the Author” section, enter your conflict of interest statement in the “Confidential to Editor” section, and submit your "Accept" recommendation.

Reviewer #2: All comments have been addressed

Reviewer #3: All comments have been addressed

2. Is the manuscript technically sound, and do the data support the conclusions?

Reviewer #2: Yes

Reviewer #3: Yes

3. Has the statistical analysis been performed appropriately and rigorously? 

Reviewer #2: Yes

Reviewer #3: Yes

4. Have the authors made all data underlying the findings in their manuscript fully available?

Reviewer #2: Yes

Reviewer #3: Yes

5. Is the manuscript presented in an intelligible fashion and written in standard English?

Reviewer #2: Yes

Reviewer #3: Yes

6. Review Comments to the Author

Reviewer #2: I think the article is presented well enough and interesting enough to merit publication at this point.

Reviewer #3: It is evident that the authors have made good efforts to address the reviewing comments, and I would recommend the revised paper to be considered for publication.

7. PLOS authors have the option to publish the peer review history of their article (what does this mean?). If published, this will include your full peer review and any attached files.

Reviewer #2: No

Reviewer #3: No

---

## [Editor Report · Acceptance letter]

31 Jan 2023

PONE-D-22-15831R1 

Probabilistic coherence, logical consistency, and Bayesian learning: neural language models as epistemic agents 

Dear Dr. Betz:

I'm pleased to inform you that your manuscript has been deemed suitable for publication in PLOS ONE. Congratulations! Your manuscript is now with our production department. 

Kind regards, 

on behalf of

Dr. Anu Sayal 

Academic Editor

PLOS ONE